# A haplotype-phased genome of wheat stripe rust pathogen *Puccinia striiformis* f. sp. *tritici*, race PST-130 from the Western USA

Hans Vasquez-Gross[1], Sukhwinder Kaur[2], Lynn Epstein[2], Jorge Dubcovsky[1,3]*

**1** Department of Plant Sciences, University of California, Davis, CA, United States of America, **2** Department of Plant Pathology, University of California, Davis, CA, United States of America, **3** Howard Hughes Medical Institute, Chevy Chase, MD, United States of America

* jdubcovsky@ucdavis.edu

**Data Availability Statement:** All PacBio sequencing data generated for this project can be found on SRA with this BioProject accession number PRJNA650506. Scripts, notebooks, and

## Abstract

More virulent and aggressive races of *Puccinia striiformis* f. sp. *tritici* (*Pst*), the pathogen causing wheat stripe rust, have been spreading around the world since 2000 causing large grain yield losses. A better understanding of the genome and genetic diversity of these new *Pst* races will be useful to develop new strategies to ameliorate these losses. In this study, we generated an improved genome assembly of a post-2000 virulent race from the Western USA designated as PST-130. We implemented a haplotype phasing strategy using the diploid-aware assembler, Falcon-Unzip and new long-read technology from PacBio to phase the two genomes of this dikaryotic organism. The combination of these new technologies resulted in an improved PST-130 assembly with only 151 contigs (85.4 Mb, N50 of 1.44 Mb), and a complementary assembly (haplotigs) with 458 contigs (65.9 Mb, N50 of 0.235 Mb, PRJNA650506). This new assembly improved gene predictions resulting in 228 more predicted complete genes than in the initial Illumina assembly (29,178 contigs, N50 of 5 kb). The alignment of the non-repetitive primary and haplotig contigs revealed and average of 5.22 SNP/kb, with 39.1% showing <2 SNP/kb and 15.9% >10 SNP/kb. This large divergent regions may represent introgressions of chromosome segments from more divergent *Pst* races in regions where a complete sexual cycle and recombination are possible. We hypothesize that some of the divergent regions in PST-130 may be related to the European "Warrior" race PST-DK0911 because this genome is more similar to PST-130 (3.18 SNP/kb) than to the older European race PST-104E (3.75 SNP/kb). Complete phasing of additional *Pst* genomes or sequencing individual nuclei will facilitate the tracing of the haploid genomes introduced by the new *Pst* races into local populations.

## Introduction

Wheat is an important crop for world food security, and provides approximately one-fifth of total food calories and protein consumed worldwide [1]. In order to feed an increasing human population, it is important to minimize yield losses due to pathogens. Wheat stripe rust, caused by *Puccinia striiformis* f. sp. *tritici* (*Pst*), is currently one of the most damaging diseases of

annotation files can be found on GitHub page: https://github.com/hans-vg/pst130_genome.

**Funding:** JD received funding for this study from the Agriculture and Food Research Initiative Competitive Grant 2017-67007-25939 from the United States Department of Agriculture, National Institute of Food and Agriculture (https://nifa.usda.gov/) and Howard Hughes Medical Institute (https://www.hhmi.org/). The funders had no role in study design, data collection and analysis, decision to publish, or preparation of the manuscript.

**Competing interests:** The authors have declared that no competing interests exist.

wheat [2, 3]. The more virulent and aggressive races that appeared in the Western USA around the year 2000 caused large wheat grain yield losses. After the initial introduction, numerous virulence combinations appeared in California within a few years, which overcame resistance genes present in previously resistant wheat varieties, and resulted in losses of >25% of the crop in 2003 [4]. Although the deployment of new resistance genes is helping to control these new races, a better understanding of the dynamics of the stripe rust populations is required to predict and prevent new epidemics.

*Puccinia striiformis* is able to infect some *Berberis* species where it can complete the sexual phase of its complex life cycle [5]. However, in the United States, only the asexual part of the *Pst* life cycle has been detected so far [6]. In this part of its life cycle, *Pst* infects wheat and produces urediniospores, which can be spread by wind over long distances and re-infect wheat without going through a sexual cycle [7]. During this asexual phase, *Pst* is in a dikaryotic stage (n + n), with each cell having two nuclei. This provides an additional level of complexity to the genomic analysis, and to the understanding of the dynamics of *Pst* evolution.

To make things more complex, several studies have suggested that *Pst* and other rust species have the ability to re-assort whole nuclei when different races infect a host, generating new virulence combinations [8–11]. This mechanism may explain the rapid appearance of new races after the introduction of the new virulent race PST-78 in the USA, in spite of the absence of a complete sexual cycle. It also highlights the importance of phasing the genomes of different *Pst* races to trace the evolutionary history of individual haploid genomes.

The first genomic studies in *Pst* were done in race PST-130 using Illumina short-read technology, which resulted in a very fragmented genome [12]. This first 65 Mb assembly included 29,178 contigs with an N50 of 5 kb. Four additional races (PST-21, PST-43. PST-87/7 and PST-08/21) sequenced with similar Illumina technology showed similar levels of fragmentation (43,106 to 55,502 contigs) and small N50s (2.6 to 4.0 kb) [13]. The assembly of race PST-78, using high coverage Illumina sequencing and a newer assembler resulted in an improved genome of 117 Mb with 17,295 contigs and an N50 of 17 kb [14]. A slight reduction in the number of contigs (12,528) and a small increase in N50 (18 Kb) was achieved by using a "fosmid-to-fosmid" strategy in the sequencing of the Chinese *Pst* race CY32 [15].

The initial attempts to sequence the *Pst* genome used small reads and generated a single consensus sequence without distinguishing between variants from the two haploid genomes. With the use of PacBio long reads and new assemblers [16], it became possible to generate separate contigs for maternal and paternal alleles, or in the case of the dikaryotic *Pst*, the two haploid genomes (phasing). Two additional *Pst* races, PST-104E [17] and DK0911 [18], have been sequenced using these new tools and have their genomes partially phased. PST-104E is the Australian founder pathotype of *Pst* collected in 1982 (before the appearance of the new virulent races), and belongs to the European clonal lineage PstS0. Its genome has been assembled into a primary assembly of 83 Mb (156 contigs) and a secondary or 'haplotig' assembly of 73 Mb (475 secondary haplotype contigs or haplotigs). PST-DK0911 was collected in Denmark in 2011 and belongs to the *Pst* race group known as "Warrior", which combine virulence to *Pst* resistance genes *Yr1*, *Yr2*, *Yr3*, *Yr4*, *Yr6*, *Yr7*, *Yr9*, *Yr17*, *Yr25*, *Yr32*, and *YrSP* [19]. This group is different from previous European races and similar to isolates from the near-Himalayan region, where they likely originated from sexually recombining populations [19]. The genome of this race was assembled into a primary assembly of 74.4 Mb (94 contigs) and a more fragmented 'haplotig' assembly of 52 Mb (1,176 haplotigs).

In this study, we combined PacBio, supplemented with Illumina sequencing for error correction, and the Falcon-Unzip assembler to improve significantly the genome assembly of PST-130, and to expand the number of predicted genes. More importantly, the use of the longer PacBio reads and the diploid-aware assembler allowed us to phase the majority of the two

genomes present in PST-130. We also explored the distribution of polymorphisms along the aligned non-repetitive regions of the two PST-130 genomes and their average relationships with the previously published phased genomes of PST-104E and PST-DK0911.

## Material and methods

### Growth conditions and harvesting

PST-130 is a highly virulent race that was first identified in the Western USA in 2007. PST-130 is equivalent to PSTv-69 in the new *Pst* nomenclature, which has virulence to *Yr6*, *Yr7*, *Yr8*, *Yr9*, *Yr10*, *Yr17*, *Yr27*, *Yr32*, *Yr43*, *Yr44*, and *YrExp2*; and avirulence to *Yr1*, *Yr5*, *Yr15*, *Yr24*, *YrSP*, *YrTr1*, and *Yr7*6 (YrTye) [20, 21]. To obtain high quality DNA of PST-130 for PacBio sequencing, we inoculated 21 days old plants of tetraploid wheat Kronos grown in a growth chamber (16 h light /8 h darkness, 16˚C constant temperature with 95–98 relative humidity). For inoculation, we injected ~1 g of spores with 100 mL of water within the basal portion of stem with a syringe. Ten to fourteen days after inoculation, we collected spores using a large cyclone spore collector (https://www.tallgrassproducts.com/) attached to a conventional wet/ dry vacuum. The spores were flash frozen in liquid nitrogen and saved in a -80˚C freezer until enough material was harvested for PacBio high molecular weight DNA extraction.

### High molecular weight DNA extraction and sequencing

In order to extract high quality high molecular weight DNA from the stored spores, we followed the protocol previously published in protocols.io (DOI: doi:10.17504/protocols.io. ewtbfen [22]). The sequencing library was prepared using the 20-kb BluePippin kit from Pac-Bio and sequenced on a PacBio RSII instrument at the University of California, Davis Genome Center.

### Bioinformatics methods

The bioinformatics pipeline is summarized in Fig 1, which includes assembly and phasing of the genomes using the Falcon-Unzip assembler [16], error correction using the PacBio reads, iterative polishing of the assembly using Pilon v1.23 [23] and Illumina reads generated previously [12]. The corrected assembly was used for coverage analyses and identification of mitochondrial and ribosomal DNA, and for the final gene annotation (Fig 1).

**Assembly and error correction.** We first tested different PacBio reads cutoffs (from 4 kb to 20 kb) to identify an optimal length using the Falcon-Unzip assembler [16]. Based on its largest initial N50 (1.5 Mb) and the largest primary contig (7 Mb), we selected a length of 12 kb. Since *Pst* cells have two nuclei (dikaryon), we implemented a haplotype phasing strategy using the Unzip method in the Falcon-Unzip assembler [16]. The last step of the Falcon-Unzip pipeline was completed by running Quiver to correct assembly errors using the PacBio reads.

We aligned the contigs and haplotigs using the MUMmer v4 suite [24], and then divided the primary assembly into primary contigs with aligned haplotigs and those without haplotigs (henceforth designated as "orphan" contigs). To test if some of the orphan contigs could fill some of the gaps in the primary contigs, we aligned the orphan contigs to the subset of primary contigs with haplotigs using the "MUMmer" program from the suite and the -mum -b -c options.

To visualize the alignments in an oriented dot plot, we aligned unmasked haplotigs to the corresponding unmasked primary contigs using the "NUCmer" program (part of the MUMmer suite). We assessed phasing completeness using '-b 200 -c 65 –maxmatch' parameters.

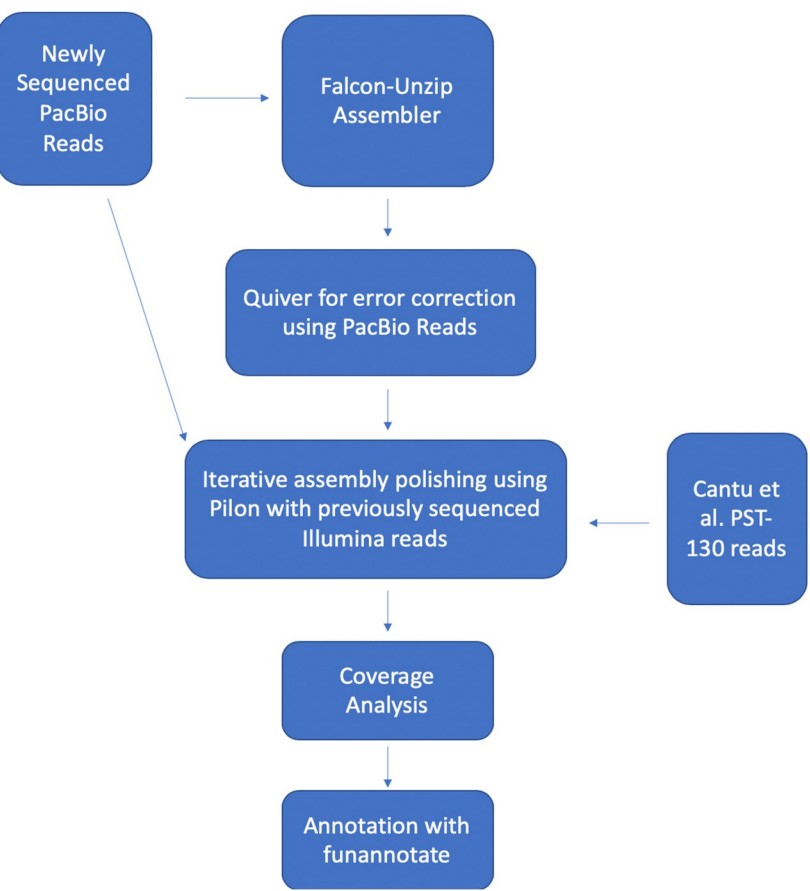

**Fig 1. Overview of the bioinformatics pipeline.** Bioinformatics pipeline used to assemble, correct and annotate the PST-130 genome.

Afterwards, we used the resulting NUCmer output delta alignment files to create an oriented dot plot using Assemblytics [25].

We performed a second round of error correction using Pilon v1.23 [23] for both the primary and haplotig assemblies, which involved multiple iterative rounds using both Illumina and PacBio reads aligned to each of the previously error-corrected reference assembly. The alignments for the Illumina reads were made using BBMap v38.79 [26] and a merged reference including both primary contigs and haplotigs. The rationale of using a combined reference was to allow the Illumina short-reads to align to their correct location in the primary or its corresponding haplotigs. Likewise, the PacBio reads were aligned to the reference assembly with Minimap2-2.17 [27] using the '-ax map-pb' option. We supplied both the Illumina and PacBio alignment files (BAM) as input for Pilon error correction. We then used the new error-corrected assembly as a new reference assembly for the next round of alignments of Illumina and PacBio reads. To select the optimum number of Pilon iterations, we ran the program, BUSCO v3 [24] or Benchmarking Universal Single-Copy Orthologs, which uses a curated set of genes for a specific organismal clade. BUSCO uses Hidden Markov models in order to search nucleotide sequence for full length or fragmented genes that match the core set.

**Mitochondria and repetitive ribosomal RNA genes.** To identify contigs from mitochondria and repetitive ribosomal RNA genes, we first performed a coverage analysis by aligning Illumina data from the previous PST-130 Illumina data [12] and the current PacBio reads to

the Pilon corrected PacBio assembly using BBMap followed by BLASTN [28] searches to Gen-Bank nt/nr database.

To estimate the average haploid PacBio coverage, we first mapped the PacBio reads to the combined primary contigs and haplotigs using 'minimap2 -ax map-pb' and estimated the PacBio coverage using samtools 'depth–aa' to produce a file with coverage at each site, and then used python to calculate the contig coverage. For this analysis, we excluded orphans and contigs containing ribosomal genes. Finally, we estimated the number of copies of each of the ribosomal gene contigs per haploid genome, by dividing their individual coverage by the estimated average haploid PacBio coverage. This number was then multiplied by the estimated number of ribosomal genes per contig to estimate the total number of ribosomal genes per haploid genome.

**Comparisons among genomes.** To perform comparisons between the haploid genomes within a single race, we first split up the reference files by primary contig and associated haplotigs. We used "nucmer" to create a delta alignment file, which was then used for input into Assemblytics. To visualize the distribution of the PST-130 genome regions aligned at different levels of identity, we generated a histogram showing the proportion of the genome aligned at 0.5 SNP/kb intervals. Separately, we calculated the proportion of the aligned PST-130 genomes that was affected by structural changes using Assemblytics and a python inotebook (GitHub page https://github.com/hans-vg/pst130_genome/blob/master/notebooks/Pst130_v5_assemblyetics_analysisV2.ipynb)

To compare the average levels of divergence among the three *Pst* races with phased genomes, we used the Assemblytics tables generated from the MUMmer alignments for pairwise comparisons. We first tried to compare the combined primary contigs and haplotigs from one race with those from the other race, but a large proportion of the comparisons were classified as repetitive because of the duplication in the two genomes. Therefore, we performed the four possible comparisons separately and then combine the UNIQUE regions for the four analyses. We used PST-104E as reference in its comparisons with PST-130 and PST-DK0911, and PST-130 as reference in the comparison between PST-130 and. PST-DK0911. In each comparison, the reference primary contig was aligned separately with the primary contigs and haplotigs of the second species, and then the process was repeated with the reference haplotig. The UNIQUE pairwise comparisons from the four alignments were combined to calculate a weighted average level of divergence between races (weighted by the length of the paired segment).

**Repeats masking and gene annotation.** To annotate the final PST-130 assembly, we used the funannotate [29] tool, which is built specifically for fungi genomes. We first downloaded a fungi specific set of repetitive elements from RepBase v25.01 (https://www.girinst.org/repbase/), and used the funannotate pipeline to run RepeatMasker and RepeatModeler to soft-mask the repetitive regions of the genome.

Following masking, we downloaded a custom set of 68,752 CDs and proteins from previous *Pst* studies including races PST-104E [17], PST-130 [12], and PST-78 [14] for gene annotation. We predicted genes using a suite of tools including Augustus [30, 31], Genemark [32], Glimmerhmm [33], and SNAP [34], which are part of the funannotate pipeline. To assess genome and annotation completeness, we used the set of universal single copy genes described in the tool BUSCO [35]. We downloaded a fungi specific gene set from Dr. Zdobnov's Lab website (https://busco.ezlab.org) for Basidiomycota (version ODB9) and we used it to run BUSCO.

After protein predictions were made, we annotated them using InterProScan5 [36], which searches Gene Ontology [37] and PFAM [38] databases. Additionally, we ran antiSMASH [39], which is specifically designed for fungi, to predict secondary metabolism gene clusters,

and eggnog mapper [40, 41], which searches CAZyme [42], UniprotKB [43], and SignalP [44] databases.

## Results

We sequenced seven PacBio SMRT cells, which after filtering yielded 447,873 reads with an N50 read length average of 21 kb. This resulted in 6,192.8 Mb of filtered sequence from the PacBio reads. This amount of sequencing was close to our target of 6,500 Mb, based on the previously published genome size of PST-130 (65 Mb) and a targeted coverage of ~100X (50X per haploid genome).

### Assembly and error correction

Phasing using the Unzip method in Falcon-Unzip resulted in a primary assembly directly from the PacBio long read technology of 88.7 Mb, including 189 contigs with an N50 of 1.5 Mb. The secondary assembly was 68.4 Mb long and included 538 contigs with an N50 of 230 kb. Since these contigs are comprised of alternative haplotypes from the primary assembly, we will be referring to them hereafter as 'haplotigs'. After correcting with Quiver, the primary assembly had 164 contigs with the same total assembly size of 88 Mb and an N50 of 1.5 Mb, and the haplotig assembly had a reduced number of contigs (458), reduced assembly size (66.4 Mb), and increased N50 (235 kb).

By aligning the orphan primary contigs to the subset of primary contigs with haplotigs using "MUMmer", we identified four orphan contigs that aligned to primary contigs along >90% of their length and another six where the alignment included at least 75% of the orphan contig. These 10 orphan primary contigs were converted to associated haplotigs using the Falcon-unzip naming convention. This brought our primary assembly to 154 contigs at 85.5 Mb (same N50 = 1.4 Mb) and our haplotig assembly to 468 contigs at 66.2 Mb (same N50 = 232 kb).

The alignment of the unmasked haplotigs to the corresponding unmasked primary contigs are presented in an oriented dot plot in Fig 2. The combined analyses of all the alignments resulted in a haplotig coverage of 54.9 Mb or 63% of the primary assembly at ~95.8% percent identity. The other 37% of the primary assembly includes unique regions from both genomes, which cannot be phased, and regions that were too divergent to be considered haplotigs by the parameters used in our alignment.

After running the Pilon error correction iterative process eight times, we saw a 3.6-fold reduction in SNPs (15,477 to 4,271) and a 9-fold reduction in indels (from 15,262 to 1,685) documenting the value of these error correction steps (Table 1). However, since Pilon can overcorrect the assemblies, we ran BUSCO after each Pilon iteration, to assess the change in completeness using the Basidiomycota database (Table 1). The first three Pilon iterations resulted in increases in the number of BUSCO complete genes, but that number remained the same after the fourth Pilon correction and started to decrease after the fifth Pilon iteration (Table 1). We decided to use the error corrected reference assembly generated after three Pilon iterations to avoid overcorrection of the assembly. The custom python script used to generate the Pilon results after each iteration (Table 1) is available on the project's GitHub page (https://github.com/hans-vg/pst130_genome).

### Identification of mitochondrial and ribosomal DNA and calculation of nuclear genome size

The PacBio coverage analysis described in Material and methods revealed 16 contigs with coverages significantly higher than the average. Twelve of the contigs had highly significant hits to

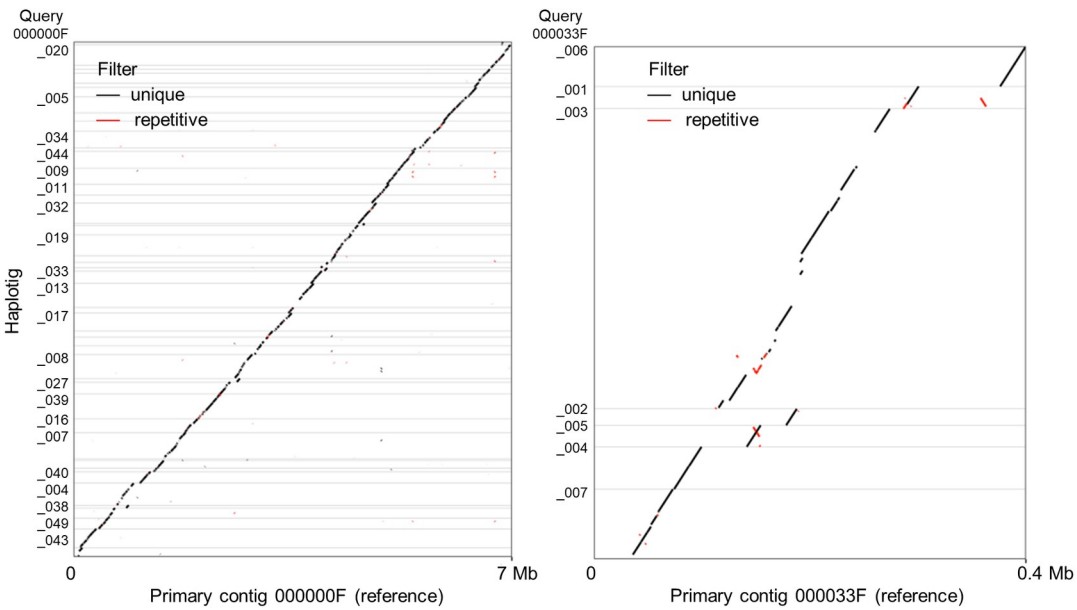

**Fig 2. Examples of haplotigs aligned to their corresponding primary contig references.** (A) The largest primary contig 00000F aligned to haplotigs shows great continuity in phasing. (B) Primary contig 000033F exhibits more repetitive regions, inversion, and gaps when compared to the haplotigs. Insertions or deletions in one of the compared sequences cause shifts in the position of the diagonal, whereas inversions cause a 90˚ rotation in the orientation.

mitochondrial genomes (E = 0) from *Puccinia* or other fungi, and among them two seem to represent full-length mitochondrial assemblies. The first one was 78 Kb, and 96% of its length aligned at 97.82% and 97.56% identity with mitochondria sequences from *Pst* races DK0911 (MN746374.1) and CYR32 (MH891489.1), respectively. The second contig was 73 Kb long, and 99% of its length was highly similar to mitochondrial assemblies from *Pst* races DK0911 (98.56%) and CYR32 (97.56%). Additionally, we downloaded the 79 kb PST-78 mitochondrial genome (AJIL01000001.1), which had the highest BLAST identity to PST-130 (99.68%) but a query coverage of only 75%. We deposited the largest mitochondrial contig from PST-130 in GenBank (BioProject PRJNA650506).

The other four repetitive contigs (000135F, 000135F_001, 000198F_001, and 000045F_001) showed highly significant BLASTN hits (E = 0, Identity = 92.23 to 93.13% identity) to ribosomal RNA contig KF792096.1 from *Puccinia* cf. *psidii* AE-2014. The KF792096.1 contig is 6,326 bp long and includes the complete 5.8S ribosomal RNA gene and the two internal transcribed spacers, and partial sequences of the 18S and 28S ribosomal RNA genes. In addition to

**Table 1. Number of changes for each Pilon correction step using Illumina and PacBio reads.**

| Type | 1st Pilon correction | 2nd Pilon correction | 3rd Pilon correction | 4th Pilon correction | 5th Pilon correction |
|---|---|---|---|---|---|
| Number of SNPs | 15,477 | 7,486 | 4,271 | 2,753 | 1,750 |
| Number of indels | 15,262 | 4,722 | 1,685 | 1,018 | 581 |
| Number of segmental changes | 879 | 250 | 121 | 58 | 43 |
| Total number of changes | 31,618 | 12,458 | 6,077 | 3,829 | 2,374 |
| BUSCO Complete | 85.1% | 85.8% | 86.6% | 86.6% | 86.5% |
| BUSCO Fragmented | 2.4% | 2.4% | 2.2% | 2.3% | 2.3% |
| BUSCO Missing | 12.5% | 11.4% | 11.2% | 11.1% | 11.2% |

**Table 2. Estimated number of ribosomal genes per haploid genome and estimated length of collapsed contigs using an average PacBio coverage of 28.8 reads per haploid genome.**

| PacBio Coverage | Name | length | Contig copy No. | Genes per contig | Estimated genes No. | Collapsed length bp[1] |
|---|---|---|---|---|---|---|
| 37.5 | 000045F (partial) | (50 of 629 kb) | 1 | 6 | 6 | 0 |
| 307.7 | 000045F_001 | 73,320 | 10.7 | 5 | 53.5 | 710,033 |
| 46.9 | 000198F_001 | 7,661 | 1.6 | 1 | 1.6 | 4,810 |
| 1520.5 | 000135F_001 | 44,063 | 52.8 | 5 | 264 | 2,282,200 |
| 167.98 | 000135F | 71,920 | 5.8 | 7 | 40.6 | 347,570 |
| | | 196,964 | 72 | | 366 | 3,344,613 |

[1] Excluding the copy already included in the genome assembly.

the four repetitive contigs, the primary contig 000045F (629 kb) showed a region of ~50 kb aligned to 000045F_001 that also include ribosomal genes.

To estimate the number of ribosomal genes in PST-130, we first estimated the approximate number of ribosomal genes in each contig based on the BLASTN analysis (Table 2) and then multiplied that number for the estimated number of haploid copies of each of these contigs. Excluding orphans and the contigs containing ribosomal or mitochondrial genes, none of the remaining contigs (68) or haplotigs (456) showed coverages >55x, and only six showed coverages >40x. The weighted average (weighted by contig length) was estimated to be 28.8 reads. By dividing the PacBio coverage of each ribosomal contig by 28.8, we estimated that the combined primary contigs and haplotigs have approximately 366 ribosomal genes (Table 2).

This analysis also helped us to calculate the length of the genome that we missed due to the collapsed ribosomal contigs. To do this, we multiplied the estimated number of haploid copies for each contig by its length, and subtracted the one copy length that was already accounted for in the calculation of the total principal contig and haplotigs. These calculations revealed that an additional 3.343 Mb should be added to the calculation of the genome size to account for the collapsed ribosomal contigs.

After the removal of the mitochondrial contigs, the final primary assembly includes 151 contigs (including two with ribosomal genes) with a total length of 85.4 Mb at an N50 of 1.44 Mb. Eighty nine percent of these contigs (134) are longer than 50 kb. The haplotig assembly includes 459 contigs (including three with ribosomal genes) with a total length of 65.9 Mb at a N50 of 235 Kb. Most of these haplotigs (332) are longer than 50 kb. The distribution of the

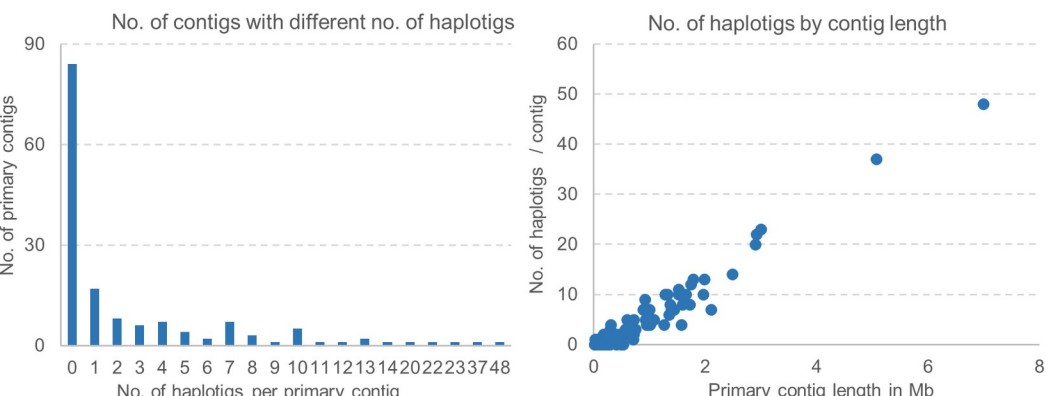

**Fig 3. Contigs and haplotigs.** (A) Frequency distribution of the number of haplotigs matching a primary contig (82 primary contigs have 0 haplotigs). (B) Correlation between primary contig size and number of aligned haplotigs.

number of haplotigs per primary contig is presented in Fig 3 together with the correlation between primary contig size and number of aligned haplotigs. The PST-130 genome has been deposited in GenBank as PRJNA650506 (https://www.ncbi.nlm.nih.gov/sra/PRJNA650506).

Among the 151 primary contigs, 82 have no haplotigs (9.559 Mb) and are considered orphans. Orphans can originate from regions present in one genome and absent in the other, from very divergent regions that the program fails to align, and also from almost identical contigs and haplotigs that the phasing programs collapse into one contig (henceforth "collapsed orphans", which is similar to the collapsed ribosomal genes described above). To identify the collapsed orphans, we used their PacBio orphan coverage based on the average haploid coverage (28.8) described above. Since collapsed contig should have twice the "haploid coverage", we estimated that collapsed orphans should have an average coverage of 57.6 and non-collapsed orphans an average coverage of 28.8 (midpoint coverage 43.2). To avoid assignment errors for contigs that were too close to the midpoint coverage, we did not assign contigs within a 10 Mb exclusion zone between 38.2x and 48.2x (43.2x ± 5.0x coverage). We sorted the 82 orphans by coverage and excluded two clones (144,159 bp) with coverages inside the exclusion zone. Among the other orphans (9.414 Mb), 97% showed a coverage below 38.2 and were considered not collapsed, whereas 3% (5 orphans = 284,799) had a coverage higher than 48.2 and were considered collapsed.

By adding the estimated collapsed orphans (284,799 bp) and collapsed ribosomal contigs (3,344,613 bp) to the primary contigs (85,402,726 bp) and haplotigs (65,887,760 bp), we estimated that the size of the two genomes is 154,919,898 bp (excluding mitochondrial DNA). If we divide this number by two, we can estimate an average haploid genome size of 77,459,949 Mb. We will refer to these two numbers hereafter to as the "corrected" total and haploid genome sizes. The uncorrected values (without the estimated collapsed contigs) are 151,290,486 for the two genomes combined and 75,645,243 for the average haploid genome size.

## Comparisons between the two PST-130 genomes

The phasing of the genomes allowed us to explore the distribution of the levels of divergence along the PST-130 genome and to estimate the average level of divergence. Using the UNIQUE hits (excluding repetitive) from the Assemblytics table generated from the MUMmer alignments, we calculated that the two PST-130 genomes have an average of 5.22 SNP/kb between the aligned primary contigs (p) and haplotigs (h) and the data is summarized in Table 3, which include similar data for the phased genomes of PST-104E (5.59 SNP/kb) and PST-DK0911 (1.96 SNP/kb). These values correlate well ($R = 0.98$) with previously reported polymorphism levels between the two genomes of each race based on the mapping of Illumina reads to their respective genome references [13, 17, 18].

To visualize the proportion of the PST-130 genomes aligned at different levels of divergence, we generated a histogram using 0.5 SNP/kb intervals (Fig 4A). We generated similar distributions for PST-DK0911 (Fig 4B) and PST-104E (Fig 4C) based on MUMmer / Assemblytics results recalculated from the published primary contigs and haplotigs [17, 18]. PST-DK0911 (Fig 4B) showed a rapid and continuous decrease of the percent identity at increasing levels of divergence, with 69.8% of the genome showing less than 2 SNP/kb and only 2.6% of the aligned genomes showing very high levels of divergences (>10 SNP/kb, Table 3). The distributions are very different in PST-130 and PST-104E, which showed a lower proportion of the aligned genomes with less than 2 SNP/kb (39.1 and 33.6% respectively), and a higher proportion of the aligned genomes with more than 10 SNP/kb (15.9 and 17.3% respectively), relative to PST-DK0911.

**Table 3. Percent divergence between the two genomes of the *Pst* races PST-130, PST-104E, and PST-DK0911.**

| | PST-130 | PST-104E | DK0911 |
|---|---|---|---|
| Aligned UNIQUE regions p *vs*. h (bp)[1] | 45,011,155 bp | 52,181,318 bp | 43,685,915 bp |
| MUMmer p *vs*. h (SNP/kb) | 5.22 SNP/kb | 5.59 SNP/kb | 1.96 SNP/kb |
| Illumina (SNP/kb)[2] | 5.4 SNP/kb | 5.0 SNP/kb | 1.6 SNP/kb |
| % of genome less 2 SNP/kb | 39.1% | 33.6% | 69.8% |
| % of genome more 10 SNP/kb | 15.9% | 17.3% | 2.6% |
| PST-130 *vs* other genomes SNP/kb | - | 3.03 SNP/kb | 3.18 SNP/kb |
| % of genome less 2 SNP/kb | | 59.4% | 59.4% |
| % of genome more 10 SNP/kb | | 6.5% | 7.3% |
| PST104E *vs* DK0911 SNP/kb | - | - | 3.75 SNP/kb |
| % of genome less 2 SNP/kb | | | 52.5% |
| % of genome more 10 SNP/kb | | | 9.8% |

[1] p = primary contig, h = haplotig.

[2] Data from [13, 17, 18].

A weighted average was calculated from the aligned UNIQUE regions by MUMmer/Assemblytics (aligned regions identified as repetitive in MUMmer were excluded for the analyses.) The % of the aligned unique regions with high (>10 SNP/kb) and low (< 2 SNP/kb) levels of divergences are also indicated.

In addition to the SNP, we found that 6.52% of the aligned regions (75.86 Mb) were involved in deletions, insertions, repeat contractions and expansions, and tandem contractions and expansions. The distribution of these structural changes among different size classes is presented in Table 4.

## Comparisons among phased PST genomes

The pair-wise comparisons of the UNIQUE regions among the three *Pst* races with phased genomes are presented in Table 3. PST-130 showed similar average levels of divergence with PST-104E (3.03 SNP/kb) and PST-DK0911 (3.18 SNP/kb), but a larger divergence was detected between PST-104E and PST-DK0911 (3.75 SNP/kb). This was reflected in a higher proportion of regions with less than 2 SNP/kb (59.4%) in the two PST-130 comparisons than in the PST-104E - PST-DK0911 comparison (52.5%). Similarly, the PST-130 comparisons showed a lower proportion of aligned regions with more than 10 SNP/kb (6.5 with PST-104E and 7.3% with PST-DK0911) than the comparison between PST-104E and PST-DK0911 (9.8% >10 SNP/kb, Table 3).

To visualize the level of colinearity and the presence of structural changes between PST-130 and PST-104E, we aligned their primary assemblies with mummer, generated a delta alignment file including the uniquely aligned regions, and used this file as input into Assemblytics. A dot plot of Assemblytics filtered alignments showed a high level of colinearity along the 60.2 Mb aligned sequences, although indels, inversions and translocation were also evident (Fig 5).

## Gene and repetitive region annotation

After the three iterative rounds of error correction with Pilon, the percentage of complete BUSCO genes in the primary contigs in DNA mode was 82.0% (8.3% missing and 9.7% fragmented) and in the haplotigs 66.4% (9.8% fragmented and 23.8% missing). After gene and protein prediction, we combined the primary and haplotigs proteins and reran the BUSCO analysis in protein mode. With the combined dataset, the BUSCO completeness score jumped to 92.6%, with only 3.4% missing and 4% fragmented proteins. To make a valid comparison between PST-130 and previously published *Pst* genomes, we re-ran the BUSCO analysis in

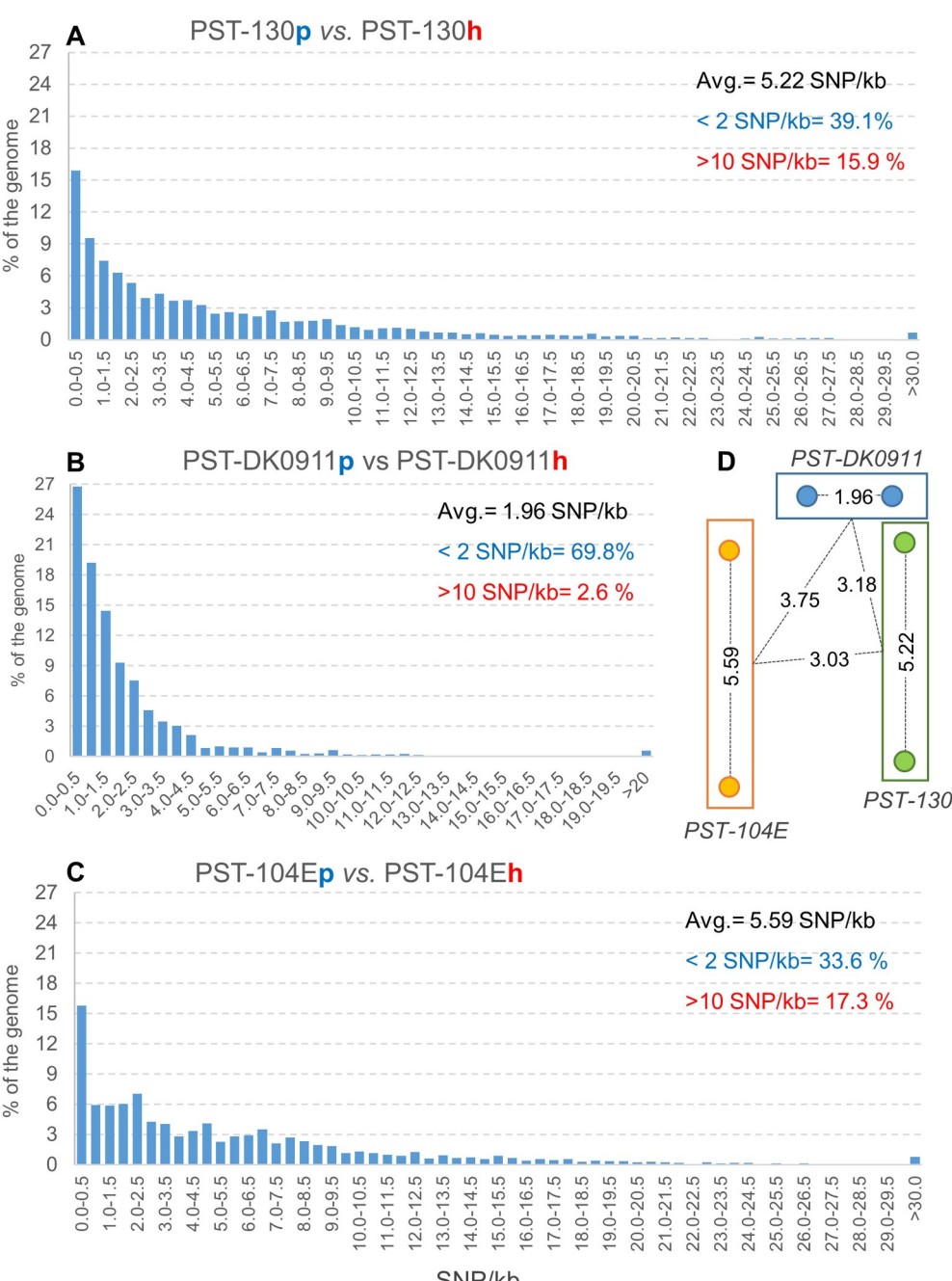

**Fig 4. Levels of divergence (in SNP/KB) between primary contigs and haplotigs and among races.** Distribution of regions with different levels of divergence in (A) PST-130, (B) PST-DK0911 and (C) PST-104E. The % of aligned regions was calculated from the UNIQUE hits (excluded repetitive) in MUMmer /Assemblytics Tables as a proportion of the total aligned regions. Regions >30 SNP/kb in PST-130 and PST-104E and >20 SNP/kb in PST-DK0911 were collapsed in the last interval. (D) Graphic summary of the levels of divergence (SNP/kb) between the two aligned genomes within each race (inside the rectangles) and the average divergence among races (among rectangles). Distances are proportional to the levels of divergence and circles represent nuclei (haploid genomes).

these genomes using their protein sets with the same Basidiomycota database used in this study (ODB9, Fig 6).

The soft-masking of the repetitive regions of the genome with RepeatMasker and Repeat-Modeler resulted in masking 24.2% of the primary assembly and 21.6% of the haplotig

**Table 4. Structural changes between primary and secondary assemblies.**

| 10 Kb | Count | Total bp | % 1–10 bp | % 10–50 bp | % 50–500 bp | % 500–10000 bp | % aligned |
|---|---|---|---|---|---|---|---|
| Deletion | 100,757 | 660,680 | 0.17 | 0.03 | 0.05 | 0.61 | 0.87% |
| Insertion | 117,364 | 1,242,023 | 0.2 | 0.03 | 0.06 | 1.35 | 1.64% |
| Repeat contraction | 539 | 1,248,464 | 0 | 0 | 0.04 | 1.61 | 1.65% |
| Repeat expansion | 505 | 1,237,682 | 0 | 0 | 0.04 | 1.59 | 1.63% |
| Tandem contraction | 48 | 121,667 | 0 | 0 | 0.01 | 0.15 | 0.16% |
| Tandem expansion | 133 | 439,006 | 0 | 0 | 0.01 | 0.57 | 0.58% |
| Total No. and bp | 219,346 | 4,949,522 | 0.37 | 0.06 | 0.21 | 5.88 | 6.52% |
| Total aligned | | 75,861,112 | | | | | |

assembly (weighted averages 23%). Following masking and using the suite of funannotate tools, we predicted 18,421 mRNA transcripts for 17,881 proteins in the primary assembly and 14,699 mRNA transcripts for 14,173 proteins in the haplotig assembly. Using the program OrthoFinder, we found 10,950 primary proteins that were orthologous to 10,978 haplotig proteins.

After protein predictions were made, we assigned functional annotations to 41.1% of the predicted proteins for the primary contig using InterProScan5 [36]. Additionally, we predicted secondary metabolism gene clusters using antiSMASH [39], and found 6 clusters, 8

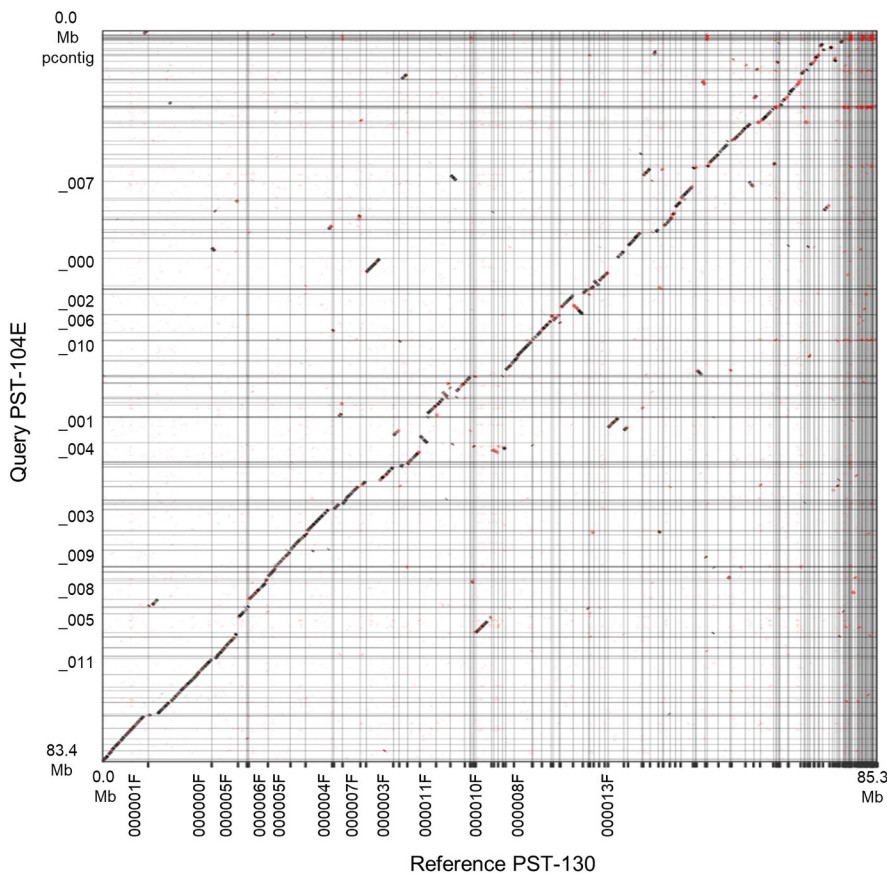

**Fig 5. Dot plot of Assemblytics filtered alignment.** The X axis reference is PST-130 and the Y axis Query is PST-104E primary assemblies. The main diagonal indicates an overall good colinearity between the two genomes. However, frequent indels, translocations and inversions (in red) are also evident.

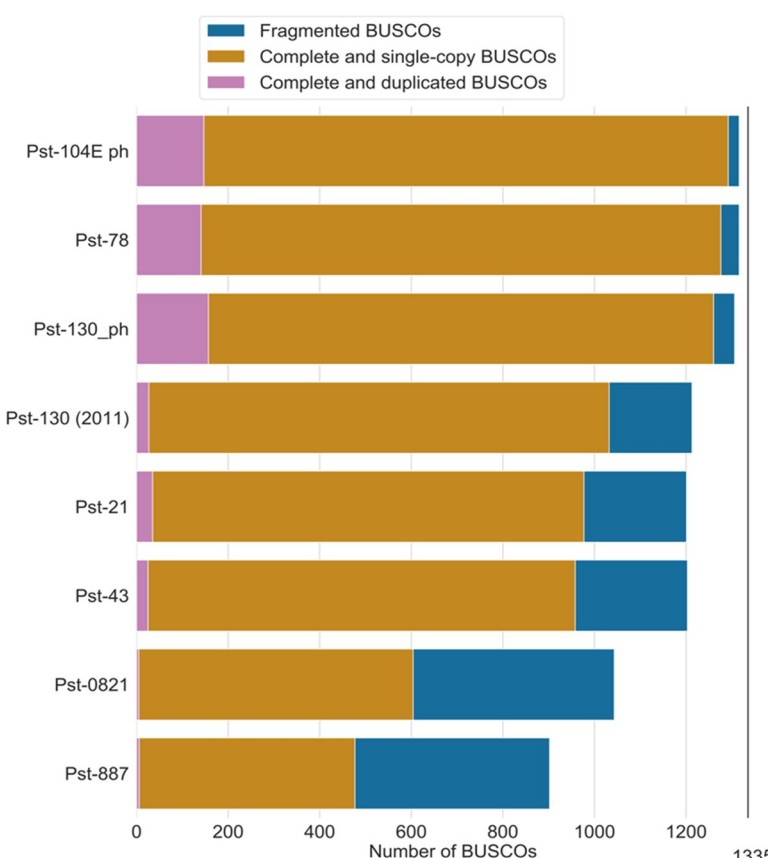

**Fig 6. Comparative BUSCO's results for different *Pst* genome assemblies.** For the phased genome, the BUSCO analysis combined the primary contigs (p) and haplotigs (h). If the BUSCO result for p was single copy, it was counted as single copy even if it was present in both p and h as single copy.

biosynthetic enzymes, and 3 secondary metabolism Clusters of Orthologous Groups (smCOGs). A final analysis using eggnog mapper [40, 41] predicted 2,084 secreted proteins. After all the functional annotations, 7,056 proteins from the primary contigs (38.3%) remained without any functional annotations. We deposited the annotations files for both the primary and haplotig assembly on our project GitHub page (https://github.com/hans-vg/pst130_genome).

# Discussion

## Comparison of PST-130 assembly with other phased *Pst* genome assemblies

The new genome assembly of the PST-130 genome based on Illumina corrected PacBio long reads represents a significant improvement over the previously published assembly of PST-130 using only Illumina short reads [12]. The new primary assembly is more contiguous with only 151 contigs and an N50 of 1.44 Mb, which represents a >250-fold increase relative to the N50 of the first PST-130 genome (5.1 kb, 29,178 contigs). In addition, the new assembly (contigs + haplotigs) showed a significant increase in complete BUSCO genes (1260) compared to the previous assembly (1032). Taken together, these results demonstrate that the new PST-130 assembly represents a significant increase in genome coverage and in contiguity relative to the initial assembly.

**Table 5. Comparison between the three first phased *Pst* genomes.** For PST-130 these are the uncorrected values without the estimated collapsed orphans and ribosomal contigs.

| | PST-130 | PST-104E | PST-DK0911 |
|---|---|---|---|
| **Primary contig** | | | |
| Length | 85.40 Mb | 83.3 Mb | 74.4 Mb |
| No. of contigs | 151 | 156 | 94 |
| N50 | 1.44 Mb | 1.3 Mb | 1.54 Mb |
| Largest contig | 6.99 Mb | 3.1 Mb | 4.5 Mb |
| Shortest contig | 16.1 kb | 21.9 kb | 20.2 kb |
| **Haplotig** | | | |
| Length (with 45F_1 repeat) | 65.89 Mb | 73.5 Mb | 52.1 Mb |
| No. of haplotigs | 459 | 475 | 1176 |
| N50 | 0.235 Mb | 0.48 Mb | 0.09 Mb |
| **(Primary contigs + haplotigs)/2** | 75.65 Mb[1] | 78.4 Mb | 74.4 Mb[2] |
| % repeats | 23.0%[3] | 53–54% | 56% |
| No. of proteins in primary | 17,881 | 15,928 | 15,070 |
| No. of proteins in haplotigs | 14,173 | 14,321 | 10,870 |
| % BUSCO genes[4] | 97.8% | 98.2% | 98.1% |

[1] With the addition of the collapsed orphans and ribosomal contigs the estimate would be 77.460 Mb.

[2] This is the corrected size estimate after accounting for collapsed orphans. In PST-DK0911 most orphans are collapsed. If we estimate the orphan contigs as (74.4–52.1 = 22.3) and add this value to the total, the average haploid genome size can be estimated as (74.4 + 52.1 + 22.3) / 2 = 74.4 Mb, which is the same as estimating the genome size from the primary contigs only. The uncorrected PST-DK0911 value would be (74.4 + 52.1) / 2 = 63.3 Mb.

[3] A different method to predict RE was used than in PST-104E and PST-DK0911.

[4] Including both complete and partial genes.

In PST-130, both the primary and haplotig assemblies showed a GC content of 44.4%, which is very similar to values reported in PST-104E (44.4%) [17], PST-CY32 (44.8%) [15], PST(93–210) and PST-2 K-41 (both 44.4%) [45]. More divergent values have been reported for *Pst* genome sizes, but some of these differences may be explained by different degrees of phasing and/or by collapsed contigs. For PST-130, we first estimated that the average haploid genome size was 75.645 Mb, but that value increased to 77.460 Mb when we added the collapsed orphan and ribosomal contigs (Table 5).

This estimate is similar to the haploid genome size for PST-104E (78.4, Table 5). Since most of the PST-104E orphans have the same coverage as the phased primary contigs and haplotigs (1x haploid coverage) [17], no adjustment for collapsed orphans is likely necessary. An independent estimate of the haploid genome of PST-104E using GenomeScope and 30-mers in two Illumina short-read data sets showed an estimate haploid genome size of 68 to 71 Mb [17], which is smaller than the estimate based on the primary contigs and haplotigs sizes.

If we calculate the average haploid genome size for PST-DK0911 by adding its primary contigs (74.4 Mb) and haplotigs (52.1 Mb), and dividing the total by two (63.0 Mb), the size would be smaller than sizes of the other two phased genomes calculated in the same way (Table 5). Since in PST-DK0911 most orphan contigs displayed a 2x haploid coverage, the main contigs provide a better estimate of the genome size [18]. This is equivalent to assume that the 22.3 Mb difference between the primary contigs and haplotigs in PST-DK0911 are all collapsed orphans ((74.4 + 52.1 + 22.3)/2 = 74.4) and is close to the average haploid genome sizes of PST-130 and PST-104E (Table 5). The higher proportion of collapsed orphans in PST-DK0911 is the expected result of the higher similarity between its two genomes (1.96 SNP/kb) compared with

the more divergent haploid genomes in PST-104E and PST-130 (5.0 and 5.2 SNP/kb, respectively).

The number of primary contigs and N50 values of PST-130 is similar to those reported for PST-104E [17] and PST-DK0911 [18] (Table 5). However, the number of haplotigs is much larger in PST-DK0911 (1,176) than in the other two races (458–475). This higher fragmentation of the haplotig contigs in PST-DK0911 is likely the result of the higher similarity between its two genomes, and the higher proportion of collapsed orphans.

## Comparison between the two haploid genomes in phased genomes

By mapping the Illumina reads to the previously assembled PST-130 reference Cantu et al. [13] detected, on average, 5.4 SNP/kb between the two PST-130 genomes [13]. This number is similar to what was reported for other four *Pst* races in the same study (5.0 to 7.6 SNP/kb) [13], and for PST-78 (6.0 SNP/kb) [14] and PST-104E (5.0 SNP/kb) [17] in separate studies.

For the three races with phased genomes, we made an independent calculation of the levels of divergences between the two haploid genomes by comparing the aligned primary contigs and haplotigs (Table 3). The correlation between the percent divergence estimated by this method and the remapping of Illumina reads to the assembly was very high ($R = 0.98$), suggesting that both methods provide good estimates of the levels of divergence. However, this correlation is based only on three pairs of data and should be consider as a preliminary result that needs to be expanded and validated. Also, these are partially phased genomes and it would be good to re-evaluate once fully phased chromosome level assemblies become available.

The calculation of divergence values from the MUMmer/Assemblytics alignments used in this study has some potential pitfalls that need to be taken into consideration. The alignment of paralogous repeats can result in the overestimation of the levels of divergence. To minimize this effect, we based our calculation only in the alignments identified in the MUMmer/Assemblytics analysis as UNIQUE. Another confounding effect that can result in an overestimation of the levels of divergence is the exclusion of the collapsed orphans from the calculation. This is not a serious problem in PST-130 because only 3% of the orphan contigs are collapsed. Even if we add 0.285 Mb from the collapsed orphan contigs with 0 SNP/kb to our calculation, the average divergences between the aligned primary contigs and haplotigs for PST-130 would be minimal (from 5.22 SNP/kb to 5.19 SNP/kb if collapsed orphans are included). However, the exclusion of collapsed orphans with 0 SNPs from the calculation of the average divergence values using the MUMmer/Assemblytics in PST-DK0911, where the collapsed orphans are abundant [18], may explain our higher divergence estimates (1.96 SNP/kb) relative to the values estimated by Illumina mapping (1.6 SNP/kb) for this race [18].

In addition to their value in the calculation of the average divergence between haploid genomes, the MUMmer/Assemblytics alignments provided a glimpse of the distribution of divergence values along the genome. These comparisons revealed different profiles in PST-DK0911 compared with PST-130 and PST-104E (Table 3 and Fig 4). Although these differences are likely influenced by the different average levels of divergence between the haploid genomes in these three races (Table 3), the profiles seem to differ more than in their mean values. In PST-DK0911, 69.8% of the UNIQUE aligned regions showed less than 2 SNP/kb, and that proportion decayed rapidly and uniformly, with only 2.6% of the aligned regions showing >10 SNP/kb (Table 3 and Fig 4). By contrast, the regions with less than 2 SNP/kb were smaller in PST-130 (39.1%) and PST-104E (34.6%), and small secondary peaks were apparent at higher levels of divergence, resulting in relatively large regions with high levels of divergence in both PST-130 (15.9%) and PST-104E (17.3%) (Fig 3). We hypothesize that these large regions with high divergence levels (>10 SNP/kb) may represent introgressions of chromosome segments

from divergent *Pst* races, in regions where *Pst* can complete its sexual cycle. For comparison, the average level of divergence between *Pst* (race PST-93-210) and *Puccinia striiformis* f. s. *hordei* (*Psh*, race PSH-93TX-2) is 7.7 SNP/kb [45], which suggest the possibility that the high divergent regions in PST-130 could have also originated from different *P. striiformis* forma specialis.

In addition to the SNP, the dot-plots of the alignments between primary contigs and haplotigs revealed the existence of frequent insertions, deletions and inversions in PST-130 (Fig 2). The quantification of these structural changes revealed that 6.52% of the 75.681 aligned Mb (REPETITIVE + UNIQUE) were affected by one of the structural rearrangement categories summarized in Table 4. A similar proportion of the aligned genome affected by structural variation was reported for PST-104E (6.39%, [17]), but the percentage was much lower in PST-DK0911 (2.66%, [18]). The structural divergence values showed a high correlation with the divergence estimates based on SNPs using both the MUMmer/Assemblytics alignments (*R* = 0.993) or Illumina reads mapped to the assembly (*R* = 0.998), suggesting that they all reflect real difference in the levels of divergence between haploid genomes in the three races.

The proportion of the genome affected by structural changes is more than an order of magnitude higher than the proportion affected by SNPs in all three *Pst* races, likely because most of the individual structural changes affect large regions (Table 4, 500–10,000 bp). However, when we used the number of structural changes in PST-130 instead of the bp affected, the number of structural changes per kb (2.89) was more similar to the number of SNP per kb (5.22) than the comparison of affected bp. As expected for the *Pst* repetitive genomes, a large proportion of the structural changes includes expansions or contractions of tandem repeats (Table 4).

## Comparison among *Pst* races with phased genomes

Since both primary contigs and haplotigs are a mixture of two haploid genomes, the comparisons among phased genomes from different races represent an average divergences between the two genomes in one race and the two genomes in the other race (Fig 4D). One additional limitation for the calculation of divergence values between races from the MUMmer/Assemblytics alignments is that the combined divergence result is affected by which genome is selected as reference, particularly if one of the genomes is highly fragmented and includes many collapsed contigs. For that reason we did not use PST-DK0911 as reference in the calculations of pairwise divergences between races. The similarity of the two haploid genomes of PST-DK0911 (<2 SNP/Kb), on the other hand, facilitated the interpretation of its comparison with the other two races. PST-130 showed an average divergence with PST-DK0911 of 3.18 SNP/kb, which is less than the average divergence between the two PST-130 haploid genomes (5.22 SNP/kb). We interpret this result as indirect evidence that one of the PST-130 haploid genomes was on average more similar to PST-DK0911 that the other one.

To test this possibility, we manually explored 4.82 Mb in the MUMmer UNIQUE table for regions were both the PST-130 primary contig (p) and haplotig (h) aligned to the same region of PST-DK0911. Within this 4.82 Mb, we identified 1.78 Mb where the divergence values of the corresponding PST-130 primary and haplotig contigs where at least 4-fold different. The weighted average of the group with the low divergence values was 1.16 SNP/kb, whereas the value for the group with the high divergence value was 7.92 SNP/kb. This result supported the hypothesis that some regions of one of the PST-130 haploid genomes were similar to PST-DK0911, whereas the orthologous region in the other haploid genome was very divergent.

However, this was not uniform across the genome, since we also identified 3.03 Mb where the primary contig and the corresponding haplotig from PST-130 showed similar levels of

divergence with PST-DK0911 (average 1.83-fold difference). These differences are consistent with the uneven distribution of divergence values between the two PST-130 haploid genomes (Fig 4A). Based on the different comparisons presented in Fig 4, we hypothesize that PST-130 (and maybe other post-2000 races of the Western USA) may have some genomic regions in common with the "Warrior" races from Europe, at least in one of their haploid genomes [19]. It is important to remember that both the primary and haplotig contigs are a random mixture of contigs from the two haploid nuclei, and that a better comparison would require fully phased genomes.

PST-104E showed higher levels of divergence with PST-DK0911 (3.75 SNP/kb) than with PST-130 (3.03 SNP/kb). A possible interpretation of this result is that PST-130 has one haploid genome that is more related to the pre-2000 races and another genome that is more related to the post-2000 "Warrior" races, whereas the two closely related PST-DK0911 haploid genomes are both related to the "Warrior" type. This hypothesis can explain why PST-130 has similar levels of divergences to PST-DK0911 (3.18 SNP/kb) and PST-104E (3.03 SNP/kb), and the larger divergences between PST-104E (two pre-2000 haploid genomes) and PST-DK0911 (two-post-2000 haploid genomes).

The hypothesis of one pre-2000 and one post-2000 haploid genomes in PST-130 is also consistent with the rapid appearance of races with new virulence combinations in the USA after the year 2000 than prior to 2000. Between the initial PST-78 race described in 2000 and PST-147 described in 2010 [46], 70 new *Pst* physiological races (using old *Pst* nomenclature) were identified in the Western USA in spite of the absence of confirmed reports of sexual recombination in the region [47]. Somatic hybridization followed by nuclei exchanges between the new races introduced in 2000 and the local *Pst* races (without requiring sexual recombination) is a potential explanation for the previous conflicting results. Although nuclei exchanges have been reported in the literature [8–11], the impact of this mechanisms on the rapid origin of *Pst* races in the USA after the year 2000 requires further validation. A similar hypothesis has been recently proposed for the origin of the *Puccinia graminis* Ug99 lineage, which shares one haploid nucleus genotype with a much older African lineage of *Pgt*, with no recombination or chromosome re-assortment [48].

## Annotated repetitive sequences

In the first PST-130 assembly using only Illumina short reads, 17.8% of the genome was identified as autonomous and non-autonomous transposable elements (TEs, 140 families) [12]. However, the previous study also pointed out that this percentage was likely an underestimate because larger sequences from similar repeats can be assembled into common contigs when using only short reads [12]. The use of larger PacBio reads allowed us to resolve better these repetitive regions, and we found that 23.0% of the primary contigs and haplotig assembly consisted of repetitive sequences. The PacBio coverage analysis showed that most of the contigs had coverages close to the 28.8x average (= 1 x haploid genome), and only the mitochondrial and ribosomal contigs showed coverages higher than 2x. These results indicate that although a much larger proportion of PacBio reads contain repetitive regions, those reads were uniquely mapped, likely because they were longer in average than single repetitive elements. Even if a repetitive element has a large number of copies, the insertion sites are unique. So, if at least one border of a repetitive element is included in most PacBio reads, those will be mapped uniquely to the genome. These results emphasize the value of the PacBio reads to do coverage analysis and to assembly large repetitive sequences.

The annotation of repetitive sequences in this new PST-130 assembly (21.6% of the genome) is higher than the previous estimate (17.5%), suggesting that the longer reads allowed

us to resolve some of the previously collapsed repeats. However, this new estimate is still lower than the 54% of repetitive regions reported for PST-104e [17] or the 49% reported for CY32 [15]. We think that these differences are likely the result of different methodologies. Our annotation of repetitive regions was based on RepeatMasker and RepeatModeler, whereas the annotation of the repetitive regions in PST-104E and PST-DK0911 was based on the REPET pipeline [49], which includes additional types of repeats and is better with *de novo* detection of repeats.

## Gene annotation

All three partially phased *Pst* genomes show a very high proportion of BUSCO genes suggesting that these assemblies are close to complete (Table 5). The total number of identified proteins in the primary contigs and haplotigs were similar in PST-130 (32,027) and PST-104E (30,249), but were lower in PST-DK0911 (25,873). This last lower number is likely affected by the large proportion of collapsed orphans in this *Pst* race, because genes in these regions are counted once instead of twice, as in the other two races. If we correct the gene number in PST-DK0911 using a simple proportionality based on the estimated haploid genome size of 77.6 Mb (including collapsed orphans) instead of 63.0 Mb, the gene number would increase to 31,868, which is similar to the number observed in the other two races.

## Conclusion

The new assembly of PST-130 represents a significant improvement over the previous PST-130 assembly published in 2011 and is the third phased genome assembly for *Pst*, and the first one from the USA. The larger PacBio reads used in the phased genome studies facilitated the resolution of most of the repetitive regions and the generation of a more contiguous and complete genome. In this study, we also show that these phased genomes can be used to provide a glimpse of the distribution of the levels of divergence across the genome. These analyses have shed light to the relationships between the haploid genomes in the different *Pst* races and the relationships among races. The sequencing of more *Pst* phased genomes and future improvements on the phasing of complete haploid genomes (e.g. by single nuclei sequencing or by HiC libraries) will provide a more clear picture of the dynamics and relative importance of sexual recombination and swapping of complete nuclei in the evolution of new virulences in this economically important pathogen.

## Acknowledgments

The authors thank the UC Davis DNA technologies for the PacBio library prep and support.

## Author Contributions

**Conceptualization:** Jorge Dubcovsky.

**Data curation:** Hans Vasquez-Gross.

**Formal analysis:** Hans Vasquez-Gross, Jorge Dubcovsky.

**Funding acquisition:** Jorge Dubcovsky.

**Investigation:** Hans Vasquez-Gross, Sukhwinder Kaur, Lynn Epstein, Jorge Dubcovsky.

**Methodology:** Hans Vasquez-Gross, Lynn Epstein.

**Project administration:** Jorge Dubcovsky.

**Software:** Hans Vasquez-Gross.

**Supervision:** Jorge Dubcovsky.

**Visualization:** Hans Vasquez-Gross, Jorge Dubcovsky.

**Writing – original draft:** Hans Vasquez-Gross.

**Writing – review & editing:** Lynn Epstein, Jorge Dubcovsky.

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
