## [Decision Letter · Decision Letter 0]

25 Sep 2020

PONE-D-20-25928

A haplotype-phased genome of wheat stripe rust pathogen *Puccinia striiformis* f. sp. *tritici*, race PST-130 from the Western USA

PLOS ONE

Dear Dr. Dubcovsky,

Thank you for submitting your manuscript to PLOS ONE. After careful consideration, we feel that it has merit but does not fully meet PLOS ONE’s publication criteria as it currently stands. Therefore, we invite you to submit a revised version of the manuscript that addresses the points raised during the review process.

Dear Prof Dubcovsky,

I received comments from the advisers on revised version of your manuscript “A haplotype-phased genome of wheat stripe rust pathogen Puccinia striiformis f. sp. tritici, race PST-130 from the Western USA“, which you submitted to PlosONE. According to opinion of our reviewers and my personal assessment, your manuscript could be taken in consideration for publishing in PlosONE if you are willing to incorporate minor changes When preparing revised manuscript, you are asked to carefully consider the reviewer comments which can be found below, and submit a list of detailed and itemized responses to the comments.

With kind regards

Dragan

We look forward to receiving your revised manuscript.

Kind regards,

Dragan Perovic, Ph.D

Academic Editor

PLOS ONE

Journal Requirements:

Additional Editor Comments (if provided):

Dear Prof Dubcovsky,

I received comments from the advisers on revised version of your manuscript “A haplotype-phased genome of wheat stripe rust pathogen

Puccinia striiformis f. sp. tritici, race PST-130 from the Western USA“, which you submitted to PlosONE. According to opinion of our reviewers

and my personal assessment, your manuscript could be taken in consideration for publishing in PlosONE if you are willing to incorporate minor

changes When preparing revised manuscript, you are asked carefully consider the reviewer comments which can be found below, and submit

a list of detailed and itemized responses to the comments.

With kind regards

Dragan

Reviewers' comments:

Reviewer's Responses to Questions

**Comments to the Author**

1. Is the manuscript technically sound, and do the data support the conclusions?

Reviewer #1: Yes

Reviewer #2: Yes

2. Has the statistical analysis been performed appropriately and rigorously? 

Reviewer #1: Yes

Reviewer #2: Yes

3. Have the authors made all data underlying the findings in their manuscript fully available?

Reviewer #1: Yes

Reviewer #2: Yes

4. Is the manuscript presented in an intelligible fashion and written in standard English?

Reviewer #1: Yes

Reviewer #2: Yes

5. Review Comments to the Author

Reviewer #1: 1. Materials and Methods contains only about fungal growth and DNA extraction. The methods utilized in the bioinformatics analyses are merged with the results (eg. lines 148 - 153, 156 - 167, 174 - 184, 187 - 193, 281 - 284, 322 - 333, 271 - 380, etc.) need to be placed in the Materials and methods section; and the results section should present only results.

2. Lines 534 - 550: Be careful not to be too speculative about the hypothesis of one pre-2000 and one post-2000 haploid genome in Pst-130 as the supporting references presented may not strongly support the hypothesis. For example, drastic changes in virulence structure post-2000 may be associated with changes in the differential lines used to designate Pst races (Wan and Chen, 2014). In addition, Liu et al. (2017) did not discuss/confirm absence of sexual recombination in Pst as presented in the manuscript. The appropriate literature here is Wang et al. (2012); not withstanding Jin et al. (2010)

Reviewer #2: The submitted manuscript (PONE-D-20-25928) by Vasquez-Gross et al. on a haplotype-phased genome of wheat stripe rust pathogen Puccinia striiformis f. sp. tritici (Pst) race PST-130 from the western USA is a well-designed study. The studies were successful in generating an improved genome assembly of a virulent race PST-130. The strength of the studies is that the authors combined PacBio technology (especially long PacBio reads) with Illumina sequencing (for error correction) and the Falcon-Unzip assembler which significantly improved the genome assembly of PST-130 and expanded the number of predicted genes. As concluded by the authors, I agree that the sequencing of more Pst phased genomes and future improvements on the phasing of complete haploid genomes will provide a much clearer picture of relative importance of sexual recombination and swapping of complete nuclei in the evolution of new virulences in Pst. The findings are of high topical interest and will be useful for many researchers working in area of pathogen genome assembly. The MS is written clearly, and the experiments are well planned, conducted, analysed thoroughly and interpreted. The results are properly discussed and well concluded. The figures submitted were not of high resolution (at least in my download copy – they appear scanned version) and definitely needs improvement if the MS get published.

I recommend the MS for publication with some minor comments:

1) In the abstract, authors mention very virulent race. What VERY virulent means? Also, in line 87, it is mentioned ‘new more virulent’. What is more?

2) Line 105 – ‘21 days old plants of tetraploid wheat Kronos’ will read better.

3) Line 245 – Put 134 in bracket or revise the sentence.

4) Line 529-530 - PST-130 may have some genomic regions in common with the “Warrior” races (Put a reference for Warrior race)

5) Table 4 – Put ‘% aligned’ instead ‘% of aligned’

6. PLOS authors have the option to publish the peer review history of their article (what does this mean?). If published, this will include your full peer review and any attached files.

Reviewer #1: **Yes: **Belayneh Admassu Yimer

Reviewer #2: **Yes: **Dr Davinder Singh

---

## [Author Response · Author response to Decision Letter 0]

5 Oct 2020

Please see attached file 'Response to Reviewers.pdf' for a better formatted response than the one entered below

Editor recommendation

Answer: We added the doi link to protocols.io and the corresponding reference, and we rephrased the old sentence to: “In order to extract high quality high molecular weight DNA from the stored spores, we followed the protocol previously published in protocols.io (DOI: doi:10.17504/protocols.io.ewtbfen [22]).” 

[22] Schwessinger B, Rathjen JP. Extraction of high molecular weight DNA from fungal rust spores for long read sequencing. Methods Mol Biol. 2017;1659:49-57. doi: 10.1007/978-1-4939-7249-4_5. PMID: 28856640.

5. Review Comments to the Author

Reviewer #1

1. Materials and Methods contains only about fungal growth and DNA extraction. The methods utilized in the bioinformatics analyses are merged with the results (eg. lines 148 - 153, 156 - 167, 174 - 184, 187 - 193, 281 - 284, 322 - 333, 271 - 380, etc.) need to be placed in the Materials and methods section; and the results section should present only results.

Answer: As requested by the reviewer, we generated a new section designated “Bioinformatics methods” compiling all the bioinformatics methods into Material and Methods. We excluded those paragraphs from the results sections.

2. Lines 534 - 550: Be careful not to be too speculative about the hypothesis of one pre-2000 and one post-2000 haploid genome in Pst-130 as the supporting references presented may not strongly support the hypothesis. For example, drastic changes in virulence structure post-2000 may be associated with changes in the differential lines used to designate Pst races (Wan and Chen, 2014). In addition, Liu et al. (2017) did not discuss/confirm absence of sexual recombination in Pst as presented in the manuscript. The appropriate literature here is Wang et al. (2012); not withstanding Jin et al. (2010)

Answer: We agree with Reviewer #1 that the presence of a pre-2000 and a post-2000 haploid genome in PST130 is just a hypothesis and we modified the sentence in the Discussion to present it as a hypothesis rather than a suggestion: “Based on the different comparisons presented in Fig 4, we hypothesize that PST-130 (and maybe other post-2000 races of the Western USA) may have some genomic regions in common with the “Warrior” races from Europe, at least in one of their haploid genomes [19].”

We also agree with Reviewer #1 that the change of rust differentials in the determination of Pst races nomenclature introduces an additional layer of complexity. However, our comment about the number of post-2000 races is based on the original set of differential lines. Using the old PST nomenclature Wan and Chen (2014) reported an increase from race PST78 in the year 2000 to race PST147 in 2010 (70 new races!). We agree with Reviewer #1 that we were not using the correct citations for the number of race and for the absence of sexual recombination. Therefore, we rephrased the sentence to: “Between the initial PST-78 race in 2000 to PST-147 in 2010 (using the old Pst nomenclature) [45], 70 new Pst races were identified in spite of the absence of confirmed reports of sexual recombination in the region [46]. Somatic hybridization followed by nuclei exchanges between the new races introduced in 2000 and the local Pst races (without requiring sexual recombination) is a potential explanation for this phenomenon [8-11] that will require further validation.”

References [45] and [46] are now:

45. Wan AM, Chen XM. Virulence characterization of Puccinia striiformis f. sp. tritici using a new set of Yr single-gene line differentials in the United States in 2010. Plant Dis. 2014;98:1534-42. 

46. Wang MN, Chen XM. Barberry does not function as an alternate host for Puccinia striiformis f. sp tritici in the U. S. Pacific Northwest due to teliospore degradation and Barberry phenology. Plant Dis. 2015;99(11):1500-6. 

The new reference of Wang et al. 2015 specifically discusses sexual recombination of stripe rust and the limited role of Barberry in Pst sexual cycle in North America. 

Reviewer #2

The submitted manuscript (PONE-D-20-25928) by Vasquez-Gross et al. on a haplotype-phased genome of wheat stripe rust pathogen Puccinia striiformis f. sp. tritici (Pst) race PST-130 from the western USA is a well-designed study. The studies were successful in generating an improved genome assembly of a virulent race PST-130. The strength of the studies is that the authors combined PacBio technology (especially long PacBio reads) with Illumina sequencing (for error correction) and the Falcon-Unzip assembler which significantly improved the genome assembly of PST-130 and expanded the number of predicted genes. As concluded by the authors, I agree that the sequencing of more Pst phased genomes and future improvements on the phasing of complete haploid genomes will provide a much clearer picture of relative importance of sexual recombination and swapping of complete nuclei in the evolution of new virulences in Pst. The findings are of high topical interest and will be useful for many researchers working in area of pathogen genome assembly. The MS is written clearly, and the experiments are well planned, conducted, analysed thoroughly and interpreted. The results are properly discussed and well concluded. 

The figures submitted were not of high resolution (at least in my download copy – they appear scanned version) and definitely needs improvement if the MS get published.

Answer; we have images of higher resolution but when we run it to the PACE web site requested by PLOS, they were transformed into lower resolution to comply with PLOS requirements. 

I recommend the MS for publication with some minor comments:

1) In the abstract, authors mention very virulent race. What VERY virulent means? Also, in line 87, it is mentioned ‘new more virulent’. What is more?

Answer: We eliminated the world VERY from the abstract. We rephrased as “… a post-2000 virulent race from the Western USA designated as PST-130”

We also replaced line 87 and eliminated the sentence “new more virulent”. The new sentence describes more precisely the virulence profile of the Warrior races and adds a new reference [19] 

PST-DK0911 was collected in Denmark in 2011 and belongs to the Pst race group known as “Warrior”, which combine virulence to Pst resistance genes Yr1, Yr2, Yr3, Yr4, Yr6, Yr7, Yr9, Yr17, Yr25, Yr32, and YrSP [19]. This group is different from previous European races and similar to isolates from the near-Himalayan region, where they likely originated from sexually recombining populations [19].

[19] = Hovmøller MS, Walter S, Bayles RA, Hubbard A, Flath K, Sommerfeldt N, et al. Replacement of the European wheat yellow rust population by new races from the centre of diversity in the near-Himalayan region. Plant Pathol. 2016;65(3):402-11.

2) Line 105 – ‘21 days old plants of tetraploid wheat Kronos’ will read better.

Answer: Edited as requested

3) Line 245 – Put 134 in bracket or revise the sentence.

Answer: We rephrased as: “Eighty nine percent of these contigs (134) are longer than 50 kb.”

4) Line 529-530 - PST-130 may have some genomic regions in common with the “Warrior” races (Put a reference for Warrior race)

Answer: Added reference [19]

5) Table 4 – Put ‘% aligned’ instead ‘% of aligned’

Answer: Edited as requested

---

## [Decision Letter · Decision Letter 1]

29 Oct 2020

A haplotype-phased genome of wheat stripe rust pathogen *Puccinia striiformis* f. sp. *tritici*, race PST-130 from the Western USA

PONE-D-20-25928R1

Dear Dr. Dubcovsky,

We’re pleased to inform you that your manuscript has been judged scientifically suitable for publication and will be formally accepted for publication once it meets all outstanding technical requirements.

Kind regards,

Dragan Perovic, Ph.D

Academic Editor

PLOS ONE

Additional Editor Comments (optional):

Dear Prof Dubcovsky,

it is my pleasure to accept your article to be published in PlosONE.

Regards

Dragan

Reviewers' comments:

Reviewer's Responses to Questions

**Comments to the Author**

1. If the authors have adequately addressed your comments raised in a previous round of review and you feel that this manuscript is now acceptable for publication, you may indicate that here to bypass the “Comments to the Author” section, enter your conflict of interest statement in the “Confidential to Editor” section, and submit your "Accept" recommendation.

Reviewer #1: All comments have been addressed

Reviewer #2: All comments have been addressed

2. Is the manuscript technically sound, and do the data support the conclusions?

Reviewer #1: Yes

Reviewer #2: (No Response)

3. Has the statistical analysis been performed appropriately and rigorously? 

Reviewer #1: Yes

Reviewer #2: (No Response)

4. Have the authors made all data underlying the findings in their manuscript fully available?

Reviewer #1: Yes

Reviewer #2: (No Response)

5. Is the manuscript presented in an intelligible fashion and written in standard English?

Reviewer #1: Yes

Reviewer #2: (No Response)

6. Review Comments to the Author

Reviewer #1: (No Response)

Reviewer #2: (No Response)

7. PLOS authors have the option to publish the peer review history of their article (what does this mean?). If published, this will include your full peer review and any attached files.

Reviewer #1: **Yes: **Belayneh Admassu Yimer

Reviewer #2: **Yes: **Davinder Singh

---

## [Editor Report · Acceptance letter]

3 Nov 2020

PONE-D-20-25928R1 

A haplotype-phased genome of wheat stripe rust pathogen *Puccinia striiformis* f. sp. *tritici*, race PST-130 from the Western USA 

Dear Dr. Dubcovsky:

I'm pleased to inform you that your manuscript has been deemed suitable for publication in PLOS ONE. Congratulations! Your manuscript is now with our production department. 

Kind regards, 

on behalf of

Dr. Dragan Perovic 

Academic Editor

PLOS ONE